# New Insights into the Mechanisms of Chaperon-Mediated Autophagy and Implications for Kidney Diseases

**DOI:** 10.3390/cells11030406

**Published:** 2022-01-25

**Authors:** Zhen Yuan, Shuyuan Wang, Xiaoyue Tan, Dekun Wang

**Affiliations:** Pathology Department, The School of Medicine, Nankai University, 94 Weijin Road, Nankai District, Tianjin 300071, China; 1710976@mail.nankai.edu.cn (Z.Y.); ShuyuanWang1030@163.com (S.W.)

**Keywords:** chaperon-mediated autophagy, kidney diseases, lysosomal associated protein 2A, heat shock-cognate 71 kDa protein

## Abstract

Chaperone-mediated autophagy (CMA) is a separate type of lysosomal proteolysis, characterized by its selectivity of substrate proteins and direct translocation into lysosomes. Recent studies have declared the involvement of CMA in a variety of physiologic and pathologic situations involving the kidney, and it has emerged as a potential target for the treatment of kidney diseases. The role of CMA in kidney diseases is context-dependent and appears reciprocally with macroautophagy. Among the renal resident cells, the proximal tubule exhibits a high basal level of CMA activity, and restoration of CMA alleviates the aging-related tubular alternations. The level of CMA is up-regulated under conditions of oxidative stress, such as in acute kidney injury, while it is declined in chronic kidney disease and aging-related kidney diseases, leading to the accumulation of oxidized substrates. Suppressed CMA leads to the kidney hypertrophy in diabetes mellitus, and the increase of CMA contributes to the progress and chemoresistance in renal cell carcinoma. With the progress on the understanding of the cellular functions and uncovering the clinical scenario, the application of targeting CMA in the treatment of kidney diseases is expected.

## 1. Introduction

Autophagy plays an important role in balancing protein synthesis and degradation, maintaining proper cellular homeostasis [1]. Unlike the ubiquitin-proteasome system that mainly aims to degrade short-lived proteins, lysosome-dependent autophagy is mainly responsible for degradation of long-lived or aggregated proteins and damaged organelles [2]. According to different modes of cargo delivery to the lysosome, autophagy can be divided into separate types: macroautophagy, microautophagy and chaperone-mediated autophagy (CMA). Macroautophagy, also referred to as autophagy, isolates damaged proteins and organelles in double-membraned vacuoles known as autophagosomes, and then fuses with lysosomes for degradation under the regulation of the autophagy-related (*Atg*) genes [1,3]. In microautophagy, the lysosomal membrane directly enwraps cytoplasmic components into the lysosomes or endosome, accompanied by invagination of the endosomal membrane known as the intraluminal vesicles [4,5]. CMA maintains the intracellular proteostasis networks together with other forms of autophagy. When CMA is impaired, macroautophagy can be up-regulated, but cannot compensate entirely. Therefore, CMA-defective cells are more vulnerable to different stressors [6]. Moreover, this compensation tends to decrease with stress or aging [7]. Both CMA and macroautophagy can be activated by starvation, but their sequential activations play different roles in response to stressors [8]. Lysosome-associated membrane protein type 2A (LAMP2A) and the heat shock-cognate chaperone of 71 kDa (HSC70) are two key proteins required for CMA. Proteins targeted for degradation by CMA cross the lysosomal membrane one-by-one through a protein translocation system with the assistance of cytosolic chaperones such as HSC70, which recognizes and binds a KFERQ-like pentapeptide motif in the substrate protein [9,10]. The interaction between KFERQ-like motif and HSC70 is not only involved in CMA, but also participates in endosomal-microautophagy and chaperone-assisted selective autophagy, while LAMP2A only occurs in CMA [11].

Almost all mammalian cell types show a basal CMA; however, liver and kidney have the highest percentage of protein containing a KFERQ-motif [12]. Glomerular podocytes have a high level of basal macroautophagy, and CMA is the predominant form of autophagy in the tubular system [13]. The relevance of this high level of CMA activity on proximal tubule physiology and pathophysiology has been well documented previously [14]. With the advances in the technique and molecular insights on CMA, more and more evidence on the role of CMA in distinct kidney diseases is emerging. It has been proven that CMA is involved in the progression of renal cancer and chemoresistance in renal cell carcinoma (RCC) patients [15]. Under the condition of oxidation, as with acute kidney injury (AKI), CMA is up-regulated to selectively degrade misfolded, oxidized, or damaged cytosolic proteins. In chronic kidney disease (CKD) and aging-related kidney disease, the CMA is declined, leading to the accumulation of oxidized substrates [16,17,18,19]. In diabetes mellitus, the CMA is decreased and the accumulation of paired-box protein (Pax2), a substrate important for renal growth, leads to kidney hypertrophy [20]. In this review, we focus on the progress with regard to the pathogenesis and implications of CMA in the kidney diseases.

## 2. Biology of Chaperon-Mediated Autophagy

The CMA process can be roughly divided into four steps: (1) binding to HSC70 and targeting to lysosomes; (2) binding to LAMP2A and unfolding; (3) translocation into the lysosomes; (4) degradation by luminal proteases (Figure 1). Approximately 40% of proteins in mammalian proteomes contain a KFERQ-like motif and about 75% of proteins contain a KFERQ-like motif after post-translational modification (e.g., phosphorylation or acetylation) [21], making CMA a crucial mechanism for the regulation of cellular proteostasis and adaptation of challenges.

A KFERQ-like motif is first identified by cytosolic HSC70 (cyt-HSC70), and they are then targeted to the surface of lysosomes where the substrate can bind to the 12-amino-acid cytosolic tail of LAMP2A [21,22]. Subsequently, the substrate unfolds, which is necessary for translocation. The interaction drives the multimerization of LAMP2A into a 700 kDa protein complex, which is required for substrate translocation into the lysosome lumen [23]. This translocation complex acts as a transmembrane protein channel and interacts with heat shock protein 90 (HSP90) at the luminal side of the lysosomal membrane to maintain the stability of the multimeric assembly [24]. Later on, the substrates translocate into the lysosome lumen and then undergo a complete degradation by lysosomal proteases. HSC70 in lysosome lumen (lys-Hsc70) is also necessary for translocation, without which CMA cannot be performed [23]. After translocation, cyt-HSC70 is released from the translocation complex for binding new substrates to continuously maintain CMA activity, and lys-Hsc70 promotes the disassembly of LAMP2A multimerization complex into monomers to bind new substrates. Degradation of LAMP2A occurs in lipid microdomains of lysosomal membranes, executed by cathepsin A and metalloproteinase [9] (Figure 1). 

## 3. Regulation of CMA

Abnormal CMA activities have been correlated with disease occurrence, which reveals the importance of a tight regulation of CMA. Two major regulation centers, LAMP2A and HSC70, are both critical to the rates of this pathway. Since cyt-HSC70 is often in excess, LAMP2A appears to be the main target for CMA regulation [25]. In contrast to the de novo synthesis of LAMP2A, the direct changes in the stability of LAMP2A at the lysosomal membrane contributes more to CMA regulation [23]. LAMP2A stability has been shown to be regulated by cytosolic signaling pathways and some molecules at the lysosomal membrane, such as the glial fibrillary acidic protein (GFAP)/elongation factor 1α (EF1α) proteins [26] and the mammalian target of rapamycin complex 2 (mTORC2)/ serine/threonine kinase 1(Akt1)/ pleckstrin homology (PH) domain and leucine-rich repeat protein phosphatase1 (PHLPP1) axis [27]. Besides, the activation of CMA is also regulated by the acidic pH in lysosomes, which is critical for maintaining the activity of lysosomal proteases and the stability of Lys-HSC70.

### 3.1. Regulation of LAMP2A

Among the processes of CMA, substrates binding to LAMP2A is the rate-limiting step. Therefore, levels of LAMP2A at the lysosomal membrane are directly correlated with CMA activity, which is regulated by de novo synthesis, redistribution, and its degradation [25,28,29]. 

LAMP2A de novo synthesis occurs when CMA is in demand. Under oxidative conditions, the transcription factors NFE2L2 and NRF2 promote *Lamp2a* transcription [30], performing a positive regulation on oxidation-induced CMA. As a response to DNA damage, CMA is activated with an increase of LAMP2A levels, both at the protein and mRNA levels [31]. During T cell activation, transcription factor NFAT1 directly initiates the transcription of *Lamp2a* [32]. Besides, signaling through retinoic acid receptor α (RARα) has been found to have an inhibitory effect on *Lamp2a* transcription [33]. 

LAMP2A multimerization is a critical step during CMA. GFAP and EF1α are two factors involved in regulation of LAMP2A multimerization in a GTP-dependent manner [26]. Unphosphorylated GFAP stabilizes the LAMP2A multimeric complex by preventing disassembly. As a reaction to GTP, EF1α is released from the lysosomal membrane and p-GFAP preferentially binds its unphosphorylated counterpart released from LAMP2A [21]. The detachment of GFAP from LAMP2A promotes the disassembling of LAMP2A multimerization and the mobilizing of LAMP2A to specific microdomains for cleavage [26], subsequently blocking the CMA.

LAMP2A redistribution is regulated by changes in the lipid composition of the lysosomal membrane. Only LAMP2A outside of the lipid-rich domain is able to multimerize. In the state of low CMA activity, LAMP2A preferentially localizes in specific lysosomal membrane microdomains of specific lipid (enriched in glycosphingolipids and cholesterol) and protein constituents, favoring the cleavage of LAMP2A by metalloprotease and cathepsin A. Once CMA is activated, LAMP2A is excluded from lipid-enriched microdomains and turns into multimerization, mediating translocation of substrates [28]. This can explain the phenomenon that disrupting membrane cholesterol increases the CMA-mediated degradation of long-lived proteins [28].

Cathepsin A regulates LAMP2A degradation through its serine protease activity [29]. This process can be promoted by divalent cations, such as Ca2+, in different physiological conditions. In patients with galactosialidosis, detective cathepsin A leads to enhanced CMA with abnormally high levels of LAMP2A [29].

### 3.2. Regulation of Chaperones

Cyt-HSC70 is the only chaperone interacting directly with the substrate that is mainly responsible for recognizing and binding KFERQ-like motifs for degradation via CMA [22]. Indeed, cyt-HSC70 also acts as a substrate protein with targeting motifs for CMA in its amino acid sequence [34]. The other form of HSC70 (lys-HSC70) is located within the lysosome, which is essential for translocation. The blockage of CMA makes no difference on the content of HSC70 in lysosomes, suggesting that lys-HSC70 is unlikely to approach the lysosomal lumen via CMA [6]. The lysosomes lack of lys-HSC70 are incapable to perform CMA, where they can bind substrates but cannot complete translocation [35]. However, the presence of lys-HSC70 makes the CMA-inactive lysosomes sufficient for CMA. Under the condition of prolonged starvation, the number of lys-HSC70 molecules within the lysosome as well as the number of lysosomes containing lys-HSC70 are both increased [8], suggesting the increased CMA activity regulated by lys-HSC70. 

HSP90, another key chaperone, presents on both sides of the lysosomal membrane. HSP90 is necessary to guarantee the stability of LAMP2A at the lysosomal membrane while transitioning be-tween the monomeric and the multimeric membrane complexes. In the absence of HSP90, LAMP2A is shown to be more susceptible to proteolytic cleavage [23].

Besides the two major chaperones involved in CMA, HSC70 and HSP90, there are a subset of co-chaperones which are also responsible for this process, including HSP40, Bcl-2 associate athanogene 2 (BAG-1), hsc70-hsp90 organizing protein (HOP), and hsc70-interacting protein (HIP) [36].

### 3.3. Regulation of Lysosomal PH

An acidic luminal pH is a feature of lysosomes, and the acid-base balance in lysosomal lumen is of particular importance. Lysosomal cathepsins are optimally active in the acidic pH which is provided by V-ATPases. The stability of lys-HSC70 is also highly dependent on lysosomal pH, where a small increase of acidic pH within the lysosome makes it unstable [34]. Exposure of renal cell lines, including MDCK, LLCPK and HK-2 cells, to high glucose concentrations increases intracellular acidification mediated by V-ATPase [37,38]. Other conditions such as amino acid starvation or treatment with epidermal growth factor (EGF), increase V-ATPase-dependent lysosomal acidification as well, promoting proteolysis via CMA to generate more amino acids [39].

In contrast, lysosomal proteolysis can be inhibited by weak bases, such as ammonia. In acidosis, the kidney produces and excretes ammonia to remove acids from the body, keeping an acid-base balance. When ammonia enters the lysosomes, it interacts with hydrogen ions and increases lysosomal pH, and therefore suppresses CMA activity. In vivo and in vitro studies have shown that ammonia induces renal hypertrophy, especially in tubule cells [40]. Notably, it is the increase of ammonia but not acidosis itself that promotes tubular hypertrophy. With changes in lysosomes pH and cathepsin activity [40,41], ammonia decreases the abundance of LAMP2A, and leads to the accumulation of KFERQ-containing proteins. In renal tubular cells, ammonia also decreases degradation of critical transcription factor Pax2, stimulating expression of specific proteins regulating cell growth [42]. Additionally, acidosis activates early genes related to growth, therefore promoting protein synthesis. Apart from acidosis, chronic exposure to oxidative stress, as an in vitro model of aging, can promote alkalization of the lysosomal pH and thus reduce CMA activity [43].

### 3.4. Regulate Molecules of CMA

Although many regulatory mechanisms are yet to be discovered, multiple factors have been revealed to participate in the regulation of CMA, such as the GFAP/EF1α proteins and the mTORC2/Akt1/PHLPP1 axis. As mentioned above, GFAP and EF1α modulate LAMP2A assembly in a GTP-dependent manner. GFAP binding to LAMP2A stabilizes the multimeric complex, and EF1α, a GTP-binding protein, is released from GFAP in the presence of GTP [26]. The lysosome-associated mTORC2 and its kinase substrate Akt1 exert a negative regulation on CMA, whereas PHLPP1 is an activator of CMA. In the basal CMA, phosphorylation of lysosomal Akt by TORC2 increases phosphorylation of GFAP, and this leads to disassembly of the CMA translocation complex at the lysosomal membrane [27]. When CMA is induced under stress, PHLPP1 is recruited at the lysosomal membrane in a Rac1-dependent manner, and it enhances CMA through the dephosphorylation of lysosomal Akt, thus stabilizing the translocation complex [27]. There is a positive-feedback loop between mTORC2 and Akt, downstream of the insulin (INS)–phosphatidylinositol 3-kinase (PI3K)–3-phosphoinositide–dependent protein kinase 1 (PDPK1) pathway [44]. Apart from mTORC2 and PHLPP1, Akt is also modulated by INS-PI3K-PDPK1 pathway in CMA [45]. Inhibition of class-I PI3K or PDPK1 could activate CMA, while selective inhibition of class III PI3Ks cloud not [45]. Interestingly, some of these regulators also contain a KFERQ-like motif, such as GFAP, EF1α, TORC2 and Akt, suggesting that they can be regulated by CMA as well [26,27].

More and more factors were found to modulate CMA. For example, lysosomal p38 MAPK has been described to directly phosphorylate LAMP2A, activating CMA and protecting cells against stress [46]. Besides, histone deacetylase 10 (HDAC10), a member of class IIbHDACs, has been reported to deacetylate HSC70. In HDAC10 knockout cells, CMA is activated with increased LAMP2A levels and the accelerated degradation of the CMA substrate, but macroautophagy is unaffected, suggesting the involvement of HDAC10 in CMA regulation [47]. Another regulator, glucocorticoids (GC), has shown a negative regulatory effect on CMA, mediating by decreasing HSC70 in a glucocorticoid receptor-dependent manner [48]. Moreover, sirtuin 3 (SIRT3) has been reported to promote CMA to prevent lipid accumulation [49]. In HK-2 cells, long-term high-glucose culture decreases SIRT3 expression through activating Notch1/Hes1 signaling [50], consistent with the declined CMA in diabetes. 

### 3.5. Starvation

Similar to macroautophagy, starvation has been confirmed as one of the best characterized stimuli for CMA. But the activation of CMA during starvation remains much longer, which is progressively activated at increased times of starvation [8]. Under nutrient removal conditions, macroautophagy achieves maximal activity at around 6h and decreases subsequently. In this process, intracellular components are degraded unselectively via macroautophagy to provide ingredients for synthesis of essential macromolecules. However, the increase of CMA activity appears after at least 8h of starvation and maintains relative activity up to 88h [8]. Under conditions of starvation beyond 8h, unselective degradation may no longer be beneficial, which could degrade essential components for cells. Therefore, CMA becomes the predominant form of proteolysis. CMA can selectively degrade nonessential proteins and recycle amino acids for essential cell processes, in addition, it can also prevent degradation of essential proteins, such as 20S proteasome [51]. For example, several glycolytic enzymes are degraded via CMA when glycolysis is reduced during starvation. Both in vitro and in vivo experiments have proved that the activation of CMA in starvation is mainly regulated by inhibited degradation of LAMP2A, so perhaps starvation reduces intralysosomal levels of cations, but not de novo synthesis [25,29]. Besides, the long-lived Iκ-B (inhibitor of nuclear transcription factor NF-κB) is degraded by CMA during prolonged starvation, resulting in activation of NF-κB [52].

CMA also plays an important role in lipid homeostasis during starvation. Lipolysis of triglyceride or lipid droplets (LD) generates free fatty acids (FFA) to provide energy. In starved rats, the degradation of LD-associated proteins PLIN2 and PLIN3 via CMA is enhanced and precedes macroautophagy [53]. The removal of LD proteins via CMA is necessary for the subsequent lipolysis, explaining the intracellular lipid accumulation in CMA defective cells [53]. Besides, sustained lipolysis contributes to ketone body formation, which activates CMA through oxidation of substrate proteins [54]. Ketone-induced CMA occurs in tissues that do not metabolize FFAs, therefore maintaining persistent proteolysis in prolonged starvation during a predominantly lipolytic stage, which creates a relationship between lipolysis and proteolysis [55].

## 4. Implication of CMA in the Kidney Diseases

Approximately 30% of kidney proteins contain a KFERQ motif, including glycolytic enzymes, pyruvate kinase, α-microglobulin, transcription factors such as Pax2 and other substrates, all of which have shown to be relevant to kidney disease [14] (Table 1). More and more studies have suggested that CMA plays a significant role in kidney pathogenesis. For example, CMA has been found to be suppressed in diabetic mellitus, CKD, lysosomal storage disorders and obesity. In the conditions of AKI or hyaline droplet nephropathy, CMA is induced to serve as a defense for cell survival. Moreover, CMA plays a negative role in renal cancer and chemoresistance (Figure 2).

### 4.1. Diabetic Renal Hypertrophy

CMA is activated during starvation, whereas in conditions related to renal growth, CMA is down-regulated. In contrast to liver, muscle or other organs which undergo a mass loss of proteins, kidney hypertrophy occurs in certain intensely catabolic conditions, such as chronic kidney diseases, acidosis and diabetes [67,68]. Pax2 is a substrate targeted for CMA and is a transcription factor critical for renal growth. In the model of streptozotocin (STZ)-induced acute diabetes mellitus, protein synthesis in the renal cortex is increased at three days and falls to the level equal to control rats at seven days, while proteolysis is decreased for at least seven days with the accumulation of CMA substrates (Pax2), indicating that reduced CMA may be the main reason for renal hypertrophy in diabetes mellitus [20]. Furthermore, diabetes mellitus is always accompanied by enhanced expression of growth factors within the kidney [69]. The suppression of CMA in acute diabetes is more likely induced by increased growth factors in the diabetic kidney but not acid loading [20]. Several growth factors in vitro have been shown to suppress CMA activity. In a renal tubular cell, epidermal growth factor (EGF) prolonged the half-life of proteins targeted for CMA by 30% [70]. Proteolysis suppressed by EGF involves EGF-receptor tyrosine autophosphorylation, activating Ras and, in turn, activating class-I PI3K [71]. Class-I PI3K is a key regulator of protein degradation and it has many different downstream pathways, among which Akt and mTOR are two major pathways downstream of it. However, in renal tubular cells, EGF suppresses CMA and increases Pax2 abundance through the PI3K/Akt pathway, but does not involve mTOR [72]. However, not all growth factors have an inhibiting effect on proteolysis. For example, in cultured NRK-52E cells, proteolysis including CMA was suppressed by EGF, accompanied by an increase of Pax2, but cannot be suppressed by transforming growth factor β (TGF-β) [42]. Instead of changing proteolysis, TGF-β promotes renal hypertrophy by preventing cell division [70]. Furthermore, ammonia can suppress CMA and promote renal growth under metabolic acidosis, as discussed above, but it does not affect protein synthesis, which is different from EGF and TGF-β [42]. All of these suggest that under diabetic mellitus, the increased growth factors and suppressed CMA with decreased degradation of Pax2 contribute to kidney hypertrophy.

### 4.2. Lysosomal Storage Disorders (LSDs) 

LSDs are characterized by accumulative undegraded macromolecules in cells resulting from lysosomal dysfunction [73]. Cystinosis, a lysosomal storage disorder caused by defects in *Ctns*, is associated with defects in CMA. The abnormal localization of LAMP2A, locating to Rab11-positive vesicles instead of lysosomes, was observed both in *Ctns*-deficient fibroblasts and in *Ctns^−/−^* mice [56], which results in impaired CMA. Kidney is the main target of cystinosis, and nephropathic cystinosis is associated with defective proximal tubule cell (PTC) functions. In *Ctns**^−/−^* mice PTC [74], mislocalization of LAMP2A leads to accumulation of CMA substrates and this is also shown in human PTC line with defective cystinosin expression (*Ctns*-KO PTC) [75]. LAMP2A trafficking is regulated by Rab11 and RILP, two components involved in lysosomal transportation, which are down-regulated in cystinosis. These studies suggest that defective CMA may directly affect PTC function in cystinosis [74,75]. In *Ctns^−/−^* mice PTC, CMA activators rescued LAMP2A mislocalization and increased cell survival [74], and in human *Ctns*-KO PTC, CMA activators rescued Rab11 expression and trafficking [75], suggesting that CMA up-regulation is a potential strategy for the treatment of cystinosis, independent of therapies of decreasing lysosomal overload such as cysteamine [56]. However, the mechanism of how CMA deficiency causes PTC defects remains unclear. Additionally, in mucolipidosis Type IV, another LSD caused by mutation in the TRPML1, impaired CMA with decreased levels of LAMP2A causes higher levels of oxidized proteins [57]. Together, these findings reveal the underlying mechanism of lysosomal storage disorders mediated by CMA.

### 4.3. Hyaline Droplet Nephropathy 

Hyaline droplets are the result of abnormal lysosomal accumulation of α2-microglobulin in the proximal tubules. Toxin-induced hyaline droplet nephropathy is caused by exposure to environmental chemicals and leads to severe cellular damage in kidney [58]. With increased level of LAMP2A, the toxin-induced accumulation in rats exposed to chemicals is mediated by an increased rate of direct transport into lysosome via CMA [58]. However, the resistance of α2-microglobulin to protease makes α2-microglobulin remain inside the lysosomal matrix for a long period of time, therefore leading to abnormal accumulation. Subsequently, single tubule epithelial cell necrosis is followed by sustained regenerative proliferation, and finally generates renal tubule tumors [59]. The consequent cell proliferation and carcinogenesis is presumably related to decreased CMA activity with dysfunctional lysosomes caused by abnormal accumulation. Besides exposure to toxin, hyaline droplet nephropathy can be induced when other compounds or metabolites bind to α2-microglobulin [76]. For example, several pharmaceutical agents promote the formation and accumulation of hyaline droplets in proximal tubular cells. These agents have an effect on reducing the proteolysis of α2-microglobulin in lysosomes [77], which is likely associated with decreased CMA activity.

### 4.4. Chronic Kidney Disease (CKD) and Kidney Aging

CKD is featured by a premature aging phenotype. A study from the National Health and Nutrition Examination Survey (NHANES) shows that the prevalence of CKD dramatically increased with age [78], while another study reveals that the number of people with end-stage kidney disease rises after the age of 45 [79]. CMA activity declines with aging in almost all tissues [16,17,18], favoring the accumulation of oxidized proteins and organelles such as mitochondria. Mitochondrial dysfunction is common in CKD and is a main cause of poor muscle function [19]. In addition, Humanin (HN), a mitochondria-associated peptide, has a decreased level of CKD in local muscle [19]. HN has been reported to protect cells from oxidative stress in aging-related diseases [80]. In vitro, HN directly activates CMA by enhancing substrate and HSP90 interaction, thus increasing the substrate in lysosomes binding and translocation [24]. Reduced levels of HN in skeletal muscle and the consequent decreased CMA in CKD patients may be the reason for mitochondrial dysfunction and therefore lead to an accelerated aging process [19]. 

Declined CMA activity with aging might cause the development of kidney aging. It has been shown that the abundance of LAMP2A is reduced in aging, while overexpressing LAMP2A can rescue the age-related decline of CMA, alleviating accumulation of damaged proteins and improving organ function [17]. Although the number of lysosomes active for CMA increases in old rats, it does not compensate for the decreased LAMP2A levels [16]. The age-associated alterations in lysosomal membrane lipid dynamics lead to instability and degradation of LAMP2A, which is the major cause of decreased CMA activity with age [81], rather than the changes of LAMP2A itself [82]. A previous study has shown that the rate of binding and up-taking of substrates by lysosomes are both lower in older rats [16]. However, the result that the uptaking rate decreased more significantly than binding might associated with the decreased LAMP2A levels with aging. In addition, it is a notable age-related change that alteration in post-translational protein modifications, such as oxidation and glycation [83], cannot only produce a KFERQ-like motif, but can also eliminate it. It can be a reason for the less oxidized proteins in lysosomes while there are greater levels of cytosol oxidized proteins in old rats [18]. 

### 4.5. Acute Kidney Injury (AKI)

AKI is a common disease with high mortality, which is usually caused by multiple factors such as renal ischemia-reperfusion injury (IRI) and sepsis. Under the condition of AKI, CMA is decreased to ensure cell survival. Production of reactive oxygen species (ROS) and oxidative stress are drivers of AKI-associated pathology. Oxidation not only promotes CMA substrate degradation by lysosomal proteases, but also promotes binding and translocation into the lysosome. This is different from CMA induced by nutrient deprivation to provide amino acids for the synthesis of essential proteins, which is mainly regulated by degradation and relocation of LAMP2A at lysosomal membrane, but not de novo synthesis [25]. CMA under oxidative stress is mainly induced by upregulating LAMP2A transcription [18]. Besides, the oxidative modification of substrates also contributes to the activation of CMA, allowing substrates to be more easily degraded via CMA [84].

Besides oxidative stress, ferroptosis also plays a pathophysiological role in AKI, which is mediated by the accumulation of lipid peroxides, and GPX4 (glutathione peroxidase 4, a protein containing KFEQR-like motif) is a key repressor of it. Ferroptosis increases LAMP2A levels to promote CMA, and, consequently, promotes degradation of GPX4 [60]. In the mouse model of IRI-induced AKI, renal tubular ferroptosis is enhanced through promoting the degradation of GXP4 via CMA, with the involvement of legumain [61]. After inhibiting legumain, ferroptosis and renal injury is attenuated, which suggests that legumain is a potential therapeutic target for AKI [61]. Further research is still required to gain insight into the role of CMA in AKI.

### 4.6. HFD-Mediated Kidney Injury

Obesity is a common health issue and a risk factor for type 2 diabetes mellitus. A high-fat diet (HFD) or obesity negatively regulates CMA and promotes the development of obesity-related CKD [81]. A recent study reveals that HFD inhibits CMA by inactivating Pax2, mediating the renal injury [63]. AMPK is a mediator of macroautophagy and it can be activated by epoxyeicosatrienoic acids (EETs), which are substrates for soluble epoxide hydrolase (sEH) [63]. Feeding mice with an HFD for eight weeks leads to a significant decrease of Pax2 and AMPK, but an increase of renal sEH. Inhibiting sEH alleviates HFD-mediated kidney injury partly due to the elevated AMPK and Pax2 [63]. In addition, Pax2 may upregulate AMPK transcriptionally, suggesting a crosstalk between CMA and macroautophagy. Another study that targeted pregnant women shows that, compared to normal weight, HSC70 expression in placental tissues was lower in the obese group, indicating that CMA activity is decreased in maternal obesity [85]. It has been proven that dietary lipids exhibit an inhibitory effect on CMA activity by the changing lipid composition of the lysosomal membrane, which is a feature of aging [25]. Therefore, adjusting the dietary lipid intake could prevent CDK against aging.

### 4.7. Renal Cancer

CMA is upregulated in many kinds of cancer cells and tumors, and the inhibition of CMA in those cells reduces their tumorigenic capabilities [86]. The tumor microenvironment is characterized by lack of nutrients, hypoxia, and high levels of ROS, all of which are CMA activators. Meanwhile, CMA promotes cancer through sustaining energy homeostasis, proliferation, resistance to stress and a changing tumor microenvironment [87]. On the other hand, CMA also plays an anti-oncogenic role in normal tissues and prevents malignant transformation [87]. This explains that in aging or in diseases when CMA activity is reduced, there is a higher risk of malignant transformation. However, the connection between anti-oncogenic and pro-oncogenic function of CMA remains largely unknown.

A survival analysis of RCC using clinical samples shows that RCC patients with higher expression of LAMP2A have a shorter progression-free survival [15], suggesting the negative regulation of CMA in the survival of RCC. A recent study reveals that CMA is activated in renal carcinoma cells and can promote the proliferation and invasion of renal carcinoma cells through pyruvate kinase isoform M2 (PKM2) in vitro [62], which plays a crucial role in cancer progression [88]. Sunitinib is a first-line drug for patients with metastatic advanced-stage RCC, but resistance to sunitinib is a severe threat to patient survival. In the RCC patients who received sunitinib for metastases followed by nephrectomy, the higher expression of LAMP2A suggests that LAMP2A-mediated CMA plays an important role in the resistance to sunitinib [15]. Therefore, CMA inhibitors might not only suppress proliferation and invasion of RCC, but also help overcome sunitinib resistance [15,62]. In addition, the inhibition of PKM2 could also be a potential therapy target of RCC [62]. Besides, Silibinin, a natural flavonoid from milk thistle, inhibits the development of RCC xenografts in vivo, accompanied by decreased HSC70 and LAMP2A, suggesting that the strong anti-proliferative effect of silibinin on RCC is mediated by CMA [89]. In summary, although these studies indicate new therapeutics targeting CMA, determining the signaling pathways which mediate the effect of these drugs on CMA in animal models of RCC is urgently needed prior to the beginning of human clinical trials. 

### 4.8. Other Kidney Diseases

Variants of apolipoprotein L1 gene (APOL1) are associated with progressive nondiabetic nephropathy, cardiovascular disease, as well as immune-associated renal diseases, including lupus nephritis [90,91]. CMA in the APOL1 mediated nephropathy has not been pursued yet, although activated macroautophagy has been reported in lupus mice and patients with lupus nephritis [92,93,94,95]. HSC70 and LAMP2A are overexpressed in MRL/lpr B cells, both at the surface and intracellularly [64]. Studies have shown that spliceosomal peptide P140 (sequence 131-151 of the U1-70K protein phosphorylated at Ser140) alleviates the activity of systemic lupus erythematosus in a phase II clinical trial [96,97], and decreases proteinuria in MRL/lpr lupus-prone mice [98]. Interestingly, P140 could bind with HSC70, thereby decreasing its expression in MRL/lpr splenic B cells and impairing the refolding properties [99]. P140 interferes with CMA also, in part, by diminishing the recruitment of HSP90, resulting in destabilization of LAMP2A at the lysosomal membrane. This therapeutic effect of P140 on lupus nephritis, not acting as an immunosuppressant, indicates the potential role of CMA in this disease. 

Autosomal dominant polycystic kidney disease (ADPKD) is the commonest life-threatening hereditary disease, most of which is caused by a mutation in either the PKD1 or PKD2 gene [100]. Macroautophagy has been shown to be suppressed in cell [101,102], zebra fish [103] and mouse models of PKD [65]. The renal mRNA expression of *Atg* genes, such as *Atg12*, *Atg3*, *beclin1*, and *p62*, is decreased in the PKD mouse model [104]. The role of CMA in this disease is still unknown. However, cyst formation in PKD leads to localized areas of hypoxia, as evidenced by increased HIF-1α in PKD kidneys, which is the substrate of CMA [65], and HIF-1α degradation via CMA is known as a major regulator of HIF-1 activity [105]. In addition, the activated mTORC2 signaling indicated by increases of pAKT1 Ser473 is observed in polycystic kidney [66]. As discussed above, mTORC2 exerts a negative regulation on CMA. These reasons suggest that an impaired CMA activity may exert an important effect in PDK. Further study of CMA in PKD is merited, especially as increased mTOR activity and apoptosis are features of PKD.

Macroutophagy is suppressed in the podocytes of patients with immunoglobulin A nephropathy (IgAN) [106,107], and partially mediated through activated mTORC1 [108]. It is still unknown whether CMA is involved in the pathogenesis of IgAN. As for other glomerular diseases, of which injury and loss of podocytes are leading factors, little evidence of CMA has been found. This may be attributed to the fact that macroautophagy is the predominant form of autophagy in glomerular podocytes under basal conditions [109], while CMA is the predominant form of autophagy in the basal state of the mature tubule [14].

## 5. Conclusions Remarks and Future Perspectives

CMA, a major pathway of the proteostasis network, plays important regulatory roles in kidney by selectively degrading key cellular proteins. The proximal tubule has high basal levels of CMA activity, and restoration of CMA benefits the aging-related tubular alternations. Accumulating evidence suggests that CMA is involved in the pathogenesis of many kidney diseases, and several proteins actively participating in the pathogenesis of kidney diseases bear KFERQ motifs. What is the role of CMA in the glomerulus, as well as in the renal morphogenesis and development? Is the protective effect of CMA restoration repeatable in CKD and diabetic nephropathy? How do we selectively upregulate CMA in the proximal tubule? What are the selective inhibitors or activators of CMA? A series of questions surrounding the regulation and implication of CMA in kidney diseases remain unanswered. Nevertheless, with the progress on the measurement and regulation of CMA activity in vivo, understanding the cellular functions and uncovering the clinical scenario of targeting CMA in the treatment of kidney diseases is expectable.

## Figures and Tables

**Figure 1 cells-11-00406-f001:**
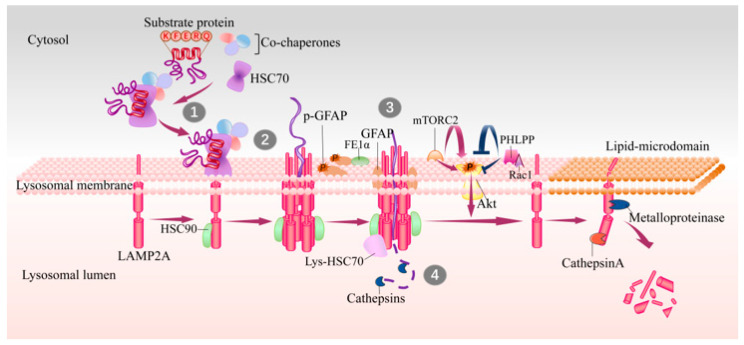
Steps and lysosomal associated proteins involved in CMA. The KFERQ-like motif in substrates are recognized by HSC70 in cytosol (step 1), and then substrate–chaperone complex binds to LAMP2A at lysosomal membrane (step 2). Substrates binding drives the multimerization of LAMP2A that mediates the translocation of substrates (step 3), and this process is assisted with lys-HSC70 and HSP90. After translocating into lysosome lumen, substrates are rapidly degraded by luminal proteases (step 4) and the translocation complex is disassembled in lipid microdomains by cathepsin A and a metalloproteinase. GFAP and EF1α are two factors involved in regulation of LAMP2A multimerization in a GTP-dependent manner. The mTORC2 complex and Akt1 exert as negative regulators on CMA, while PHLPP1 is an activator of CMA. CMA, chaperone-mediated autophagy; HSC70, heat shock cognate 71 kDa protein; GFAP, glial fibrillary acidic protein; EF1α, elongation factor1α; PHLPP1, pleckstrin homology (PH) domain and leucine-rich repeat protein phosphatase1; HSP90, heat shock protein 90; lys-HSC70, HSC70 in lysosome lumen; mTORC2, mammalian TOR complex 2.

**Figure 2 cells-11-00406-f002:**
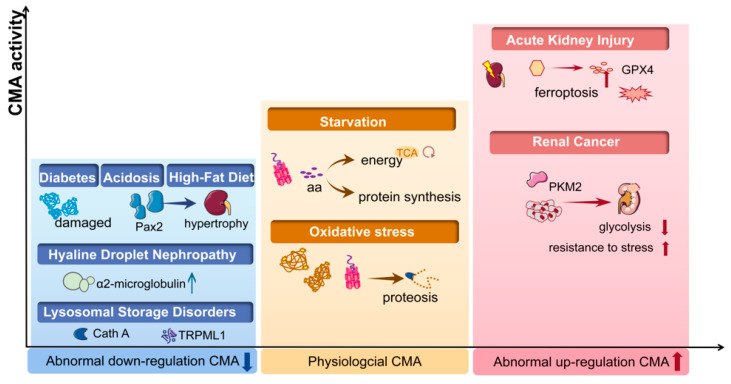
The CMA activity in kidney disease. In physiological conditions (yellow box), CMA maintains the intracellular homeostasis by degrading damaged proteins under stress and by degrading proteins to generate amino acids (aa) during starvation. Abnormal down-regulated CMA activity has been found in acidosis, diabetes, high-fat diet/obesity and lysosomal storage disorders (blue box), while abnormal CMA is up-regulated under conditions of renal cancer and acute kidney diseases (red box). CMA, chaperone-mediated autophagy; PKM2, pyruvate kinase M2; Cath A, cathepsin A; aa, amino acid; Pax2, paired box 2; TRPML1, transient receptor potential mucolipin-1; GPX4, glutathione peroxidase 4.

**Table 1 cells-11-00406-t001:** Kidney diseases and the substrates associated with defects in CMA.

Disease	Substrates	CMA Activity	Roles of CMA	Reference
Diabetes	PKM2, GAPDH	↓	The excessive PKM2, GAPDH may provide more energy for kidney hypertrophy by upregulating the glycolysis activity	[20]
	Pax2	↓	The decreased degradation of Pax2 results in kidney hypertrophy	
Cystinosis	N/A	↓	LAMP2A is mislocalized and CMA is impaired	[56]
Mucolipidosis Type IV	GAPDH, RNase A	↓	The mutation in the TRPML1 leads to MLIV and decreases CMA activity	[57]
Hyaline droplet nephropathy	α2-microglobulin	↑	Elevated proportion of direct uptake α2-microglobulin into lysosomes	[58,59]
Acute kidney injury	GPX4	↑	Ferroptosis is enchanced and increased-CMA promotes the degradation of GPX4	[60,61]
Aging	RNase A, GAPDH	↓	The decreased CMA leads to the accumulation of oxidized substrates	[16,17,18]
Cancer	PKM2	↑	CMA promotes the proliferation and invasion of renal carcinoma cells through PKM2	[62]
Starvation	nonessential proteins (e.g., glycolytic enzymes)	↑	CMA provides amino acids for the synthesis of essential proteins	[51]
	PLIN2, PLIN3	↑	Lipolysis of LD generates free fatty acids to provide energy and CMA promotes degradation of LD-associated proteins.	[53]
Acidosis	Pax2, GAPDH	↓	The decreased degradation of Pax2 results in kidney hypertrophy.	[42]
High-fat diet/Obesity	Pax2	↓	High-fat diet leads to decrease of Pax2 and AMPK and an increase of renal sEH, mediating the renal injury	[63]
Lupus nephritis	GAPDH	↑	Interfere with the endogenous autoantigen processing and loading to MHCII and lead to lower activation of autoreactive T cells.	[64]
Polycystic kidney disease	N/A	↓?	Impaired CMA may be associated with increased apoptosis and cyst growth.	[65,66]
IgAN/ Glomerulonephritis	N/A	?	N/A	

↓, Decreased; ↑, Increased; ?, uncertain currently; CMA, chaperone-mediated autophagy; PKM2, pyruvate kinase M2; GAPDH, Glyceraldehyde-3-phosphate dehydrogenase; Pax2, paired box 2; LAMP2A, lysosome-associated membrane protein type 2A; MLIV, mucolipidosis type IV; TRPML1, transient receptor potential mucolipin-1; GPX4, glutathione peroxidase 4; PLIN2, perilipin 2; PLIN3, perilipin 3; LD, lipid droplet; sEH, soluble epoxide hydrolase; MHCII, major histocompatibility complex class II.

## Data Availability

Not applicable.

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
