# Peer review of "New Insights into the Mechanisms of Chaperon-Mediated Autophagy and Implications for Kidney Diseases"

_cells, 2022, doi:10.3390/cells11030406_

Round 1

Reviewer 1 Report

Reviewer comments

The manuscript is well-written but the authors need to address the following points to improvise the manuscript

  1. What is the role of CMA in APOL1 mediated nephropathy?
  2. Mention the role of CMA in other chronic kidney diseases a) Glomerulonephritis b) ADPKD c) IgA associated nephropathy.
  3. Add relevant references for the above-mentioned kidney diseases.
  4. List the above kidney diseases in Table 1

Reviewer 2 Report

I read the review manuscript by Zhen Yuan and colleagues with interest and found it to be very informative. I just have a few comments to further improve quality, please see below:

  1. For an abstract to be more attractive, rather highlight some of the new information discussed within the manuscript, instead of just generalizing?
  2. Eliminate typos, like in the word “introduction”
  3. Check and define all terms like “CMA” from first mention
  4. Add reference(s) for the sentences “In the chronic kidney disease (CKD) and aging-related kidney…” and “In the diabetes mellitus, the CMA is decreased and the accumulation of paired-box protein…”
  5. Although it remains important for the authors to mention “we focus on the pathological role of CMA and its implication in the kidney diseases” but it should also come-up/be clear why they focus on this aspect… within the introduction. Also, are there any review papers on the topic? If none then mention that … or cite previous papers and mention how the current review is different
  6. To better guide the reader, within the introduction, better mention the core/important kidney diseases that are discussed within the manuscript in relation with CMA, and mention why those conditions…
  7. Rather add citations for all information (evidence) presented in Table 1. Also include description of terms as part of the footnote
  8. Figure captions should also describe full terms mentioned on each figure
  9. Under each subheading on the “Implication of CMA in the kidney diseases” it is better to end of by briefly discussing the state of research surrounding each condition (preclinical/clinical research).
  10. Consistently, mention if there are any therapies known to regulate CMA in order to reverse a that specific pathological state

Author Response

Please see tha attachment.

Reviewer 3 Report

It is a quite nice review. A  weak poit is that the contribution of the authors to research on the chaperon-mediated autophagy is very limited.

In numerous places it is hard to distinguish between studies on animals and in humans. This aspect should be improved.

There are several typos: 

P oor  - line 347

Patents - line 417

who plays - line 421

Sunitinib malate = sunitinib  - line 422

Many – numerous? – line 438

Author Response

Point 1: In numerous places it is hard to distinguish between studies on animals and in humans. This aspect should be improved.

Response 1: We have added the description for distinguishing studies on animals and in human.

Point 2: There are several typos: P oor  - line 347; Patents - line 417; who plays - line 421; Sunitinib malate = sunitinib  - line 422; Many – numerous? – line 438

Response 2: We have checked throughout the manuscript and corrected the typos.

Round 2

Reviewer 3 Report

No further comments